# From Stand to Forest: Woody Plant Recruitment in an Andean Restoration Project

**DOI:** 10.3390/plants13172474

**Published:** 2024-09-04

**Authors:** Marina Piquer-Doblas, Guillermo A. Correa-Londoño, Luis F. Osorio-Vélez

**Affiliations:** 1Departamento de Ciencias Forestales, Facultad de Ciencias Agrarias, Universidad Nacional de Colombia, Medellín 50034, Colombia; 2Departamento de Ciencias Agronómicas, Facultad de Ciencias Agrarias, Universidad Nacional de Colombia, Medellín 50034, Colombia

**Keywords:** natural regeneration, active restoration, mixed plantation, native species, Colombia, Andes

## Abstract

The growing deforestation of tropical forests requires the implementation of restoration actions capable of assisting the recovery of biodiversity and the functioning of these ecosystems. This research aimed to identify the environmental factors that influence the abundance and diversity of woody plant recruitment in an Andean forest restoration project in Medellin (Colombia). Data from woody plant individuals taller than 80 cm were collected in 22 plots of 200·m^−2^. The environmental factors selected were edaphic variables, plantation structure, slope, elevation, prior land use, and landscape forest cover. Generalized linear models (GLM) were used to analyze recruitment densit and Linear Mixed Models (LMM) to assess recruited species richness, diversity, and dominance. Woody plant recruitment attributes in our study area were similar to those of secondary succession in an Andean forest, but planted trees contributed little to recruitment density and diversity. While recruitment density was affected by slope, canopy closure, and landscape forest cover, recruitment diversity was influenced by physical (bulk density) and chemical (pH, aluminum, Cation Exchange Capacity) edaphic factors, planted tree diversity (species richness and composition), canopy closure, and the mortality rate of planted trees. We conclude that sites with lower mortality rates of planted trees and denser canopies enhance both recruitment density and diversity, indicating a synergy between active restoration and passive regeneration processes.

## 1. Introduction

Rates of tropical forest degradation and deforestation continue to grow worldwide [1]. In spite of the well-known effects of shrinking tropical forests, like accelerating climate change, loss of global biodiversity, and diminishing ecosystem services [2,3,4,5], more than half of the global deforestation occurs in the tropics [6,7]. This trend results in decreasing remnants of primary forests, surrounded by a complex landscape matrix of multiple land uses and secondary forests, which impacts the connectivity and resilience of primary forests [8,9,10]. In this scenario, complementing forest conservation efforts with ecological restoration actions can help to reverse and buffer tropical forest loss [11].

Ecological restoration aims to assist in the recovery of an ecosystem that has been degraded, damaged, or destroyed [12]. This discipline relies on ecological succession theories to design restoration strategies that facilitate ecosystem recovery. World tropical regions, containing the highest levels of biodiversity and ecosystem complexity, and where more than 50% of forested area consists of secondary forests, are key to understanding the successional pathways leading to primary forests [11]. However, the multiple successional theories developed during the last century still need further research to unravel the mechanisms that drive ecological succession [13].

Active restoration involves implementing actions specially designed to recover forest ecosystems, and is especially suitable to test successional theories, as it allows experimentation with multiple factors that can affect succession [13,14,15,16]. However, this potential of active restoration projects remains under-utilized due to a lack of long-term monitoring [17,18,19]. Linking ecological processes with active restoration through effective monitoring can broaden our knowledge of ecological succession in tropical forests and can contribute to our understanding of the potential of restoration for recovering the function and structure of forests.

Colombia is one of the Earth’s most diverse tropical countries. Unfortunately, it is also an accurate example of the deforestation problems described above [20]. The country’s most populated area, the Andean region, entered the XXI century having lost 69% of its original forest cover [21]. To address deforestation, an increasing number of restoration projects have been implemented in the region. However, most of them lack a good monitoring system to determine if restoration objectives are met and which ecological mechanisms are related to the successional pathways followed by the restored forests [22,23].

In this research, we conducted a woody plant recruitment study in an active restoration project of native Andean forest in Colombia. Plant recruitment is one of the most important processes in ecological succession, since it describes the spontaneous arrival and establishment of new individuals [11]. This research aims to characterize woody recruitment density and diversity to understand the mechanisms driving these attributes by relating them to multiple environmental factors: soil, planted trees structure and diversity, landscape metrics, and previous land use.

We hypothesized that woody plant recruitment density and diversity will differ across the study area in response to different environmental factors that affect recruit arrival (landscape matrix, planted tree composition, planted tree mortality, canopy cover) and establishment (soil and previous land use). In particular, we expect that sites immersed in a more forested landscape matrix, higher planted tree diversity, low planted tree mortality rates, and denser canopies will have more dense and diverse recruitment, since these conditions are favorable for seed arrival through dispersion. The forest cover landscape matrix is a proxy of key ecological mechanisms like seed dispersal or seed availability [8,9,24]. Since the recruitment process takes place in a forest plantation, we also expect recruitment features to interact with planted trees. Sites with low mortality of planted trees and denser canopies will offer more facilitation mechanisms to recruits, like an enhanced microclimate or reduced herb competition for light and nutrients [16,25,26]. Also, higher planted tree diversity is expected to support greater recruitment diversity and density, since it offers more diverse environmental conditions and can act as a source of seeds [27,28]. We also expect greater density and diversity of recruits in sites with fertile soils and with less intensive previous land use, resulting in better conditions for recruit establishment. Edaphic factors affect plant germination, establishment, and survival [11,29], so better edaphic conditions will enhance recruitment establishment. Finally, since land uses have long-lasting legacies in areas undergoing ecological succession, areas with more intensive land use will present harder conditions for recruitment establishment [29,30].

## 2. Results

### 2.1. Recruitment Characteristics

We found 1903 recruits in the 22 sampled plots, averaging 80 (±59) individuals per plot and 0.4 individuals·m^−2^ (Figure 1). Recruitment density varied greatly among plots, with a maximum of 201 individuals (0.10 individuals·m^−2^) and a minimum of 11 individuals (0.05 individuals·m^−2^). Density also varied among prior land uses. Although these differences were not significant (Kruskal-Wallis, χ^2^ = 5.60, G.L. = 2, *p*-value = 0.06), there is a marginal effect that indicated less recruitment density in plots with a prior presence of exotic pastures (*Pennisetum clandestinum* Hochst. ex Chiov.) compared with native pastures (Dunn test, Z = −2.73, *p*-value = 0.02).

For recruitment diversity, we found 131 operational taxonomic units: 99 species and 32 morphospecies (6% of total recruitment), belonging to 56 genera and 34 families (Table A1). The most abundant genera were *Miconia*, *Palicourea*, *Myrsine*, and *Verbesina*. The most abundant families were Melastomataceae, Asteraceae, Rubiaceae, and Primulaceae.

Most of the recruit diversity and abundance corresponded to native species that were not related to planted species. Only 24 recruited species (18,34% of recruit diversity) were part of planted tree species, while 67.77% of recruited individuals did not belong to planted tree species. The ten most abundant species accounted for 65.45% of recruited individuals, with endozoochory (60%) and anemochory (40%) dominating the dispersal modes (Table 1).

For Hill numbers, minimum sample coverage was 79.75%, which means that 79.75% of recruit community belongs to the 131 species and morphospecies found in this study (Table A2). Mean values of effective species were 7.42 (±3) for species richness (D_0_), 4.9 (±1.83) for species diversity (D_1_), and 3.63 (±1.42) for species dominance (D_2_). None of the three Hill numbers showed significant differences between plots or prior land use, indicating no differences in species richness, diversity, and dominance in the 22 sampled plots (Figure 2).

### 2.2. Recruiment Density and Diversity and Environmental Factors

Denser canopies, steeper slopes, and a more forested landscape matrix favored recruit density. Recruitment density was best explained by canopy cover (β = 6.17 ± 2.60), plot slope (β = 0.02 ± 0.01), forest cover proportion within a 1 km radius (β = 0.01 ± 0.01), and sand content (β = −1.54 ± 0.90) (Figure 3: Density). This model explains 65% of observed variation in data. Canopy cover, plot slope, and forest cover in 1 km present a positive and significant effect in recruitment density (*p* value < 0.05), while sand content has a negative, non-significant effect (*p* value: 0.11).

Recruit species richness was also higher in plots with denser canopies. Additionally, physical soil properties interact with chemical soil properties and with planted species richness, establishing a spectrum of habitat conditions that influences recruit species richness. Linear mixed models for species richness (D_0_) show an effect of canopy cover (β = 40.98 ± 9.78), and the interaction of bulk density with planted species richness (β = −2.94 ± 1.01) and Cation Exchange Capacity (CEC) (β = 12.24 ± 3.72) (Figure 3: D_0_). Those plots with a denser canopy contain a larger number of recruited species. The plots with high bulk density and high CEC and, conversely, low bulk density and low CEC, show greater recruitment species richness. Finally, those plots with high bulk density and less planted species, or plot with low bulk density and more planted species, also show greater recruitment species richness. The model explains 80.40% ($R_{c}^{2}$) of observed data variation. A random factor (5-km buffer) has a variance ($\sigma_{a}^{2}$) of 0.692.

Recruit species diversity (D_1_) and dominance (D_2_) are affected by similar environmental factors, being favored by sites with acid pH, low aluminum concentration, and lower mortality of planted trees (Figure 3: D_1_ and D_2_). Linear mixed models show that edaphic conditions of pH (β = −37.11 ± 12.05 for D_1_, β = −23.97 ± 10.58 for D_2_) and aluminum (β = −6.91 ± 2.10 for D_1_, β = −5.09 ± 1.89 for D_2_) had a negative, significant effect in recruitment diversity and dominance, as well as the maximum mortality rate of planted trees (β = −38.08 ± 9.85 for D_1_, β = −24.07 ± 8.81 for D_2_).

Canopy cover and planted tree composition also influenced species diversity (D_1_). Canopy cover has a significant, positive effect in recruitment diversity (β = 15.30 ± 6.47 for D_1_). The second principal coordinate (Figure A1) shows a positive, significant effect in recruitment diversity (β = 3.24 ± 1.27 for D_1_). The positive values of this axis represent plots with more abundance of planted species *Montanoa quadrangularis*, *Retrophyllum rospigliosii*, *Psidium sp.*, and *Quercus humboldtii*, while negative values are associated with plots where the species *Croton magdalenensis* is more abundant. Plots with denser canopy cover and high abundance of *M. quadrangularis*, *R. rospigliosii*, *Psidium* sp., and *Q. humboldtii* support a greater recruitment diversity. Planted species richness shows a positive, non-significant effect on recruitment dominance (*p* value: 0.07) (β = 0.19 ± 0.09 for D_2_).

The models for D_1_ and D_2_ explain 67.70% and 61.50% ($R_{c}^{2}$) of observed data variation, respectively. The random factor (5-km buffer) has a variance ($\sigma_{a}^{2}$) of 0.662 and 0.612, respectively.

## 3. Discussion

### 3.1. Recruitment Density and Environmental Factors

Our study shows that a ten-year-old restored Andean forest can achieve successional trajectories that are very similar to those found in secondary Andean forests. Recruitment density in this study is greater than density found in secondary forests of Medellin municipalities (for Diameter at Breast Height (DBH) > 10 cm, 0.2 individuals·m^−2^ versus 0.1 individuals·m^−2^ found by [37]) and in passive regeneration lowlands of Costa Rica (for DBH > 5 cm, 0.27 individuals·m^−2^ versus 0.1 individuals·m^−2^ found by [38]). However, recruitment in primary Andean forests is much denser, achieving values of 4 individuals·m^−2^ [39]. These values can be interpreted in light of the stand forest development hypothesis, in which vegetation density declines when the canopy closes, light becomes a limiting resource, and competition causes the death of many individuals, a phase known as the stand exclusion stage [13]. The less dense recruitment in secondary forests could indicate that they have already entered the stand exclusion stage, whereas our restored forest, where canopy closure has not occurred yet, could be in the previous phase, where vegetation is denser.

Sites with denser canopies and steeper slopes surrounded by a more forested landscape substantially increased recruitment density. Since canopy closure has not occurred yet, it can offer appropriate microclimatic conditions while allowing direct light to reach the understory. Facilitation effects of canopy over a recruited community have been found in other studies, and include enhanced microclimate or reducing competition with grass species [16,25,29]. At the landscape level, more forested matrices provide a greater seed pool and facilitates faunal movements, which increases seed dispersal, favoring recruitment processes [24,29,40,41]. Since animals disperse 60% of the most abundant species in this study, faunal movements play an important role for recruited community in the restored forest. The positive effect of a forest matrix in species richness recovery has been demonstrated [9]. Our study shows that a forest matrix can also affect recruitment density and suggests that the role that a landscape matrix plays in tropical forest secondary succession is broader.

Land use did not explain the differences observed in recruitment density among the plots. However, the positive effect of slope in recruitment density could be a proxy for land use intensity, since areas with steeper slopes tend to be deforested later in Colombia [42,43]. The role that land use plays in forest succession remains one of the biggest gaps in successional studies, mainly because the information about its type, intensity, and duration is often scarce [29,30]. We could not find differences in recruitment among previous land use, but the marginal negative effect of foreign pastures (*Pennisetum clandestinum*) in recruitment density suggests that land use legacies deserve a closer focus in Andean forest restoration studies.

### 3.2. Recruitment Diversity and Environmental Factors

Recruitment diversity values were also comparable to those found in other secondary forests in Colombian Andes. We found 131 species in total, a value of species richness that is equivalent to secondary forests nearby our study area: 109 species in secondary oak forests (*Quercus humboldtii*) located in Medellin [44], and 190 woody plant species in a secondary forest of Central Andean Cordillera [45]. The most abundant species, genera, and families we found are reported as common pioneer species in Andean forest succession [44,46,47], and even in Neotropical succession [11]. These results show that Andean forest restoration efforts in Medellin rural areas can be effective for recovering the diversity of secondary Andean forests.

Edaphic factors influenced the three diversity measures considered in this study. Cation Exchange Capacity (CEC) and bulk density had a positive influence on species richness through their interaction, in which species richness peaked at extremes values of both CEC and bulk density. Bulk density was directly associated with clay content, which plays a major role in soil CEC. Diversity and dominance were negatively influenced by aluminum concentration and pH. Aluminum has toxic effects on plant root development at high concentrations and low pH conditions, and increased mortality rates of planted trees in our study area [48,49]. However, low pH increased recruitment diversity and dominance, which could indicate an adaptation of recruitment species to acid soils, since they are common in the Andes [50]. Further research is needed to better understand the mechanisms that connect these edaphic factors to recruitment diversity. Exploring recruitment species composition and demographic rates could be useful for unveiling these mechanisms.

We also present facilitation effects between planted trees and recruited individuals. Canopy cover had a positive effect in species richness and diversity, probably through the same mechanisms that affect recruitment density. Planted tree maximum mortality rate had a negative effect on recruitment diversity and dominance, indicating that a loss of tree cover in plantations limited the species that can survive in the understory. Planted species composition also influenced recruitment diversity, which was greater in plots containing *Montanoa quadrangularis*, *Quercus humboldtii*, *Retrophyllum rospigliosii*, and *Psidium* sp., while the presence of *Croton magdalenensis* reduced recruitment diversity. Another study found different functional traits associated with ecohydrological processes that separated *C. magdalenensis* from *Q. humboldtii*, and *R. rospigliosii*, which suggest that planted species composition could affect recruitment community through its functional traits [51].

### 3.3. Implications for Andean Forest Restoration

In a decade of restoration efforts in tropical Andes, the resultant forest can support a recruitment community equivalent to those found in Andean secondary forests. The community is mostly composed of native species that are not related to planted trees. This indicates that forest secondary succession can progress in densely populated areas like Medellin, the second largest city of Colombia, where the landscape matrix is characterized by a great fragmentation and the coexistence of many types of land uses. In this project, forest restoration took place in former croplands and cattle ranches with native or foreign grasses. Previous land uses showed no effect for recruited community attributes. Since recruitment is a strong proxy for forest regeneration potential, it could indicate that restoration activities are suitable for these three types of land use in the tropical Andes. However, maintenance activities and early stages of active restoration projects are recommended to avoid the risk of arrested succession by weed and fern species.

Seed dispersal, especially fauna mediated dispersal, played a major role compared to seed bank or planted tree reproduction. Landscape forest cover facilitated faunal movements and increased recruitment density, so selecting areas surrounded by more forested matrices could reduce the costs of active restoration projects.

The poor contribution of planted trees to the recruited community could be explained by their young age. Long-term research will reveal if their contribution to plant recruitment increases when they reach their reproductive stage. Long-term studies of recruitment demography rates are recommended to understand interactions between planted trees and recruited individuals, since the frontier between facilitation and competition mechanisms is determinant for forest community assembly [16].

Canopy cover was the planted tree attribute which contributed the most to the facilitation effect over recruitment density and diversity. Choosing species with dense and broad canopies enhanced recruitment in restoration projects, especially in areas where cattle ranching took place and grasses can compete with young trees, allowing the implementation of cost effective restoration techniques like applied nucleation [52]. The identity of planted species matters for recruitment processes, so further research in functional traits of Andean tree species could help identify which traits enhance recruitment processes, thus accelerating restoration success.

Monitoring restoration activities in tropical Andes is a great opportunity to learn about forest succession in a biodiversity hotspot characterized by highly degraded and populated areas. We encourage long-term monitoring and experimental design of restoration techniques in order to create a synergy between ecological restoration and ecology that broadens our understanding of tropical montane forests while improving their extension and resilience.

## 4. Conclusions

This study demonstrates that one decade of active restoration techniques of Andean forests can recover woody plant recruitment density and diversity values that are equivalent to secondary Andean forests. Successional trajectories did not differ among plots or previous land use, indicating that restoration techniques can be effective in multiple deforestation contexts. Recruitment was mostly composed of native species that were not related to planted trees, but planted trees enhanced recruitment density and diversity through facilitation processes, which indicates that restoration can recover natural trajectories of secondary succession in the Andes.

## 5. Materials and Methods

### 5.1. Study Area

This study took place in rural areas of Medellin, Colombia (6°15′00.0” N 75°34′05.0” W), where an active restoration project of native Andean forest (*Mas Bosques para Medellin*) has been implemented since 2010, with more than 600 ha restored to date. This project consists of the establishment of mixed plantations of 115 species (91% native from Colombia) in areas with previous land uses (crops and cattle farming). Planted trees have been monitored biannually since 2011 across 70 permanent plots of 707 m^2^ each, in which every planted tree is tagged. We measured woody plant recruitment in an 8 m radius from the plot center (201 m^2^).

Data were collected in 2021 in the 22 oldest permanent plots, which were 10 years old when the recruitment measurement was done (Figure 4). The altitudinal range of selected plots is 2010 to 2690 MASL, with mean annual values of 14.5 °C for temperature and 2200 mm for precipitation, which places them in the wet montane forest life zone [53]. The dominant soil types in the study area are Andisols. We measured as recruitment all woody plant individuals > 80 cm height. We measured total height in cm and DBH for each recruit taller than 130 cm and identified them to the highest taxonomic level. Each recruit was marked with a unique ID for future measurements.

### 5.2. Response Variables

We selected four response variables for describing recruitment characteristics per plot: recruitment density (individuals·m^−2^), and three measures of recruitment diversity (Hill numbers D_0_, D_1_, and D_2_) [54]. Hill numbers are a parametric family of diversity indices that obey a replication principle, have units (effective number of species) that enable easy comparisons between assemblages, and have equivalences with classic diversity measures like species richness (D_0_), Shannon-Wiener entropy (D_1_), and Gini-Simpson dominance (D_2_) [54]. We standardized Hill numbers with a measure of sample completeness, specifically the minimum sample coverage found in the plots (79.75% of sample coverage), to reduce abundance bias in diversity measurements [55]. We performed all diversity measurements with *iNEXT* package in R software 4.1.2 [56,57].

### 5.3. Predictor Variables

#### 5.3.1. Edaphic Conditions

We collected five soil samples per plot and homogenized them for a fertility analysis, in which we measured pH, CEC, organic matter, phosphorus, aluminum, copper, iron, manganese, zinc, and soil texture (sand, silt and clay). We also collected one sample per plot with a metallic cylinder for measuring bulk density.

#### 5.3.2. Plantation Structure and Diversity

To relate the recruitment density and diversity to soil properties, plantation structure, and landscape metrics, we adjusted four lineal models, one for each response variable: density. We used the database with the results of planted tree monitoring in years 2015, 2017, and 2019, and calculated the following measurements for each plot: basal area per plot in 2019, average mortality rate, and maximum mortality rate (year 2017) [58], and species richness (D_0_) in 2019. Additionally, we performed a Principal Coordinate Analysis for planted species composition in 2019 following [15] (p. 4), and included the first two axes as a measure of which planted species are more abundant per plot, since species identity can play a role in recruitment density and diversity through interspecific interactions (Figure A1). Additionally, we included from this database plot elevation, slope, and previous land use, which had three categories: crops, native pastures, and exotic pastures (*Pennisetum clandestinum* Hochst. ex Chiov.). We also measured canopy cover on each plot with hemispherical photography using a Nikon Coolpix 950 and an 8-mm Sigma lens. We made five photos per plot and analyzed them with the software HemiView 2.1 to obtain mean canopy cover per plot [59].

#### 5.3.3. Landscape Metrics

We carried out a visual classification of current land cover using the Medellin orthophoto from 2021 (ESRI World Imagery, 15 m of resolution). We distinguished three types of land covers: (1) forested areas, (2) crops and pastures, and (3) urban areas. Following the analyses proposed by [9] (p. 4), we classified the land cover in a radius of 1, 2, 3, and 4 km around the plots. We then calculated landscape metrics for each radius with the software V-LATE 2.0 in ArcMap [60]. For each land cover class, we calculated total area, proportion, mean patch size, total edge, and mean patch edge. We also calculated the total number of patches, dominance, and edge density for the three land cover classes combined (landscape level analysis).

### 5.4. Statistical Analyses

To relate the recruitment density and diversity with soil properties, plantation structure, and landscape metrics, we adjusted four lineal models, one for each response variable: density, species richness, diversity, and dominance. For density, we adjusted a generalized linear model (GLM) with Poisson-distributed errors. We detected subdispersion in our standard errors, so we corrected them using a quasi-GLM where the variance is given by φ × μ, where μ is the mean, and φ the dispersion parameter [61]. For the three Hill numbers (D_0_, D_1_ and D_2_), we performed three linear mixed models. We detected spatial autocorrelation in our response variables within a 5 km radius, so we created a new variable to locate each plot in its corresponding 5 km radius and included this new variable as a random factor in our mixed models.

Since we had a large dataset comprising numerous environmental variables, we performed a preselection of the three types of explanatory variables in order to avoid multicollinearity problems. For edaphic variables, we first made a Principal Component Analysis (PCA) and included the first and second axes in our linear models, but these axes were systematically excluded from them. We excluded those edaphic variables that were strongly correlated (clay and silt with sand and bulk density), and then adjusted a saturated linear model with all edaphic variables (Equation (1)) and performed a backward selection using the stepAIC procedure from *MASS* package in R [62]. The stepAIC method selected the best model based on Akaike Information Criterion (AIC) values, but we also checked the AIC for small samples (AICc), since our number of explanatory variables was higher than the number of observations.
y = β_0_ + β_1_ logpH + β_2_ MO + β_3_ logP + β_4_ logAl + β_5_ logCEC + β_6_ arcsenA + β_7_ bulk_dens + β_8_ Fe + β_9_ Cu + β_10_ logMn + β_11_ logZn + ε(1)
where logpH: log-transformed pH, MO: organic matter (%), logP: log-transformed phosphorus, logAl: log-transformed aluminum, logCEC: log-transformed CEC, arcsenA: arcsin-transformed sand (%); bulk_dens: bulk density, Fe: iron, Cu: copper, logMn: log-transformed manganese, logZn: log-transformed zinc.

Landscape metrics were strongly correlated between them, so we selected the landscape metric that showed the strongest relationship with response variables. For this, we constructed a set of linear mixed models including one of the landscape metrics as a covariate, following [15] (p. 3). We selected the model with the lowest AICc, which was the one that included forest proportion within a radius of 1 km. We included this landscape metric along with the plantation variables and performed another linear model backward selection with stepAIC and AICc (Equation (2)).
y = β_0_ + β_1_ PCoA1 + β_2_ PCoA2 + β_3_ canopy + β_4_ BA + β_5_ mean_mort + β_6_ max_mort + β_7_ D_0__19 + β_8_ elevation + β_9_ slope + β_10_ land_use + β_11_ forest_1km + ε(2)
where PCoA1: first principal coordinate, PCoA2: second principal coordinate, canopy: canopy cover, BA: basal area in 2019, mean_mort: mean mortality rate (2015–2019), max_mort: maximum mortality rate (2017), D_0__19: planted species richness 2019, elevation: plot elevation in m, slope: plot slope in %, land_use: prior land use per plot, forest_1km: forest proportion within a 1 km radius.

Finally, we built a saturated model using the preselected edaphic variables and the preselected plantation variables, including all possible interactions between them. We performed the last backward selection with stepAIC and AICc to obtain the final model for each response variable. We also checked that all final models had a Variance Inflation Factor (VIF) lower than 10 [61]. Model assumptions were checked graphically following the method proposed by [61].

All the statistical analyses were performed using R software 4.1.2 [57]. We constructed the mixed models with the *nlme* package [63], and the GLM with the *stats* package. We standardized all variables in order to compare the magnitude of their partial regression coefficients with the *base* package. We calculated the AICc with the *qpcR* package [64].

## Figures and Tables

**Figure 1 plants-13-02474-f001:**
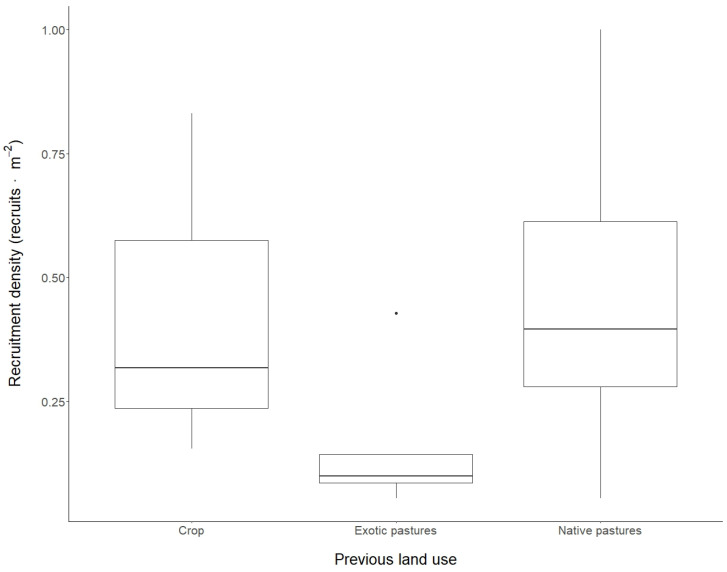
Woody plant recruitment density in the 22 sampled plots of the native Andean forest restoration project in Medellin, Colombia. Plots are grouped by previous land use: crop (three plots), exotic pastures (five plots), and native pastures (14 plots). The sampling took place in 2021, when the plots were 10 years old.

**Figure 2 plants-13-02474-f002:**
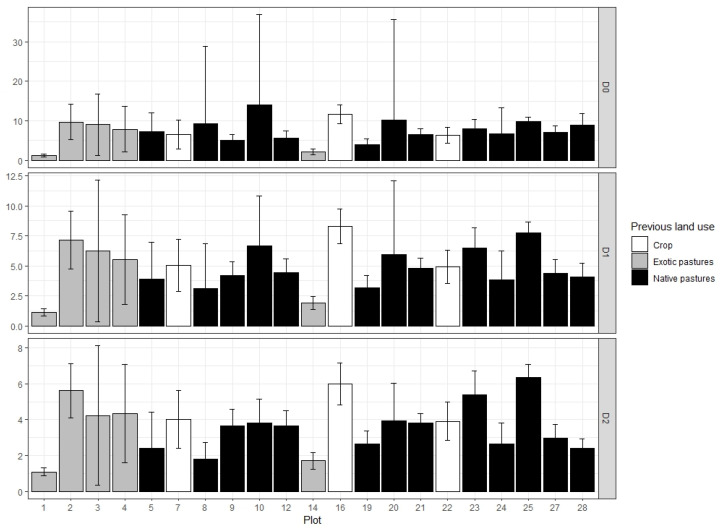
Hill numbers for woody plant recruitment in sampled plots along with 95% confidence intervals (D_0_: species richness, D_1_: species diversity, D_2_: species dominance). Values are standardized with minimum sample coverage (SC = 79.75%). Previous land use has the following color code: white for crops, grey for exotic pastures (*Pennisetum clandestinum* Hochst. ex Chiov.), and black for native pastures.

**Figure 3 plants-13-02474-f003:**
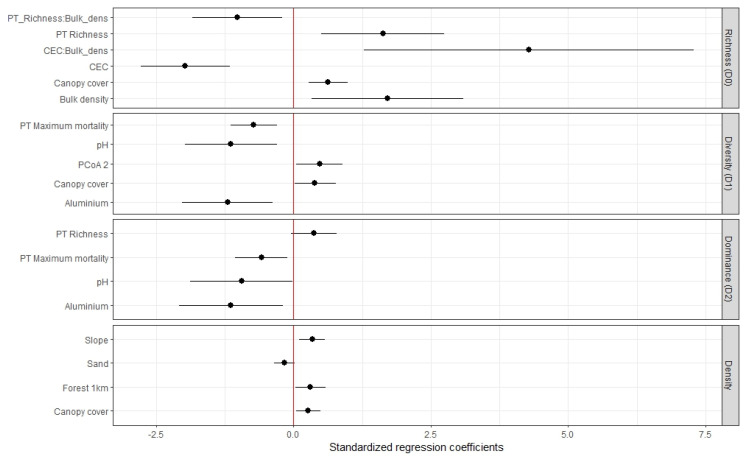
Effects of ecological factors on the density and diversity of woody plant recruitment in an Andean restoration project in Medellin, Colombia. PT: Planted Trees; CEC: Cation Exchange Capacity. Black dots represent the standardized regression coefficients with black horizontal lines showing 95% confidence intervals. Points located right to the red line (zero) indicate positive effects, while points located left of it indicate negative effects.

**Figure 4 plants-13-02474-f004:**
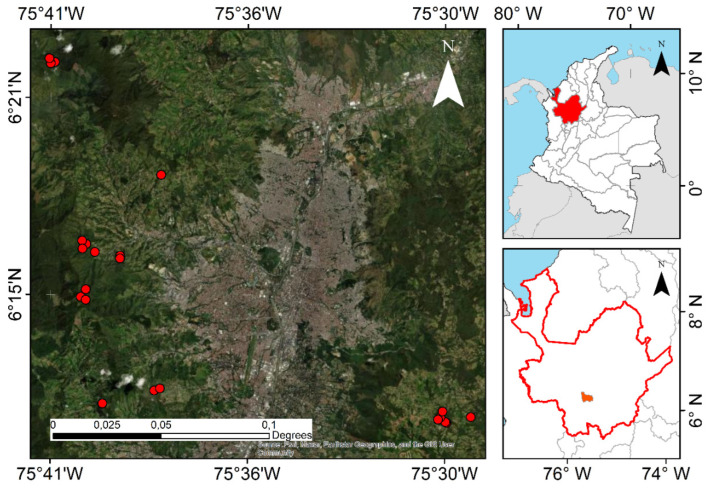
Study location in Medellin, Colombia. Red points show the location of the 22 plots where woody plant recruitment was sampled, all of them belong to an Andean forest restoration project (*Mas Bosques para Medellin*).

**Table 1 plants-13-02474-t001:** Ten most abundant species of woody plant recruitment in the 22 sampled plots of the native Andean forest restoration project in Medellin, Colombia. The sampling took place in 2021, when the plots were 10 years old. An asterisk near a species name means that species is also present among the planted trees.

Species	Number of Individuals	Presence in Plots	Dispersal Mode
*Miconia theaezans*	305	71.43%	Endozoochory [31]
*Myrsine coriacea* *	186	80.95%	Endozoochory [32]
*Verbesina helianthoides*	121	57.14%	Anemochory (achene)
*Palicourea acetosoides*	109	4.76%	Endozoochory [32]
*Montanoa quadrangularis* *	92	38.10%	Anemochory [33]
*Croton magdalenensis* *	85	57.14%	Endozoocoria [34]
*Fraxinus uhdei*	70	4.76%	Anemochory (samara)
*Palicourea thyrsiflora*	65	14.28%	Endozoochory [32]
*Viburnum undulatum*	49	42.86%	Endozoochory [35]
*Weinmannia pubescens* *	49	23.80%	Anemochory [36]

## Data Availability

The data presented in this study are openly available at https://figshare.com/articles/dataset/plantrecruitment_dataset_xlsx/26264234 (accessed on 15 July 2024).

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
