# Peer review of "From Stand to Forest: Woody Plant Recruitment in an Andean Restoration Project"

_plants, 2024, doi:10.3390/plants13172474_

Round 1
Reviewer 1 Report
Comments and Suggestions for Authors
I would like to thank the authors for the nicely written manuscript and interesting study. I have a few major comments and furthermore minor comments (see below). The major comments are 1) to test for the effect of prior land use in your regression models to make your claim that prioir land use does not have an effect on recruitment density and diversity more convincing, 2) in case you do not run a model with soil characteristics and landscape metrics together, please do so as in that way you can analyse what the relative importance is of the different independent variables (either through comparing the standardized regression coefficients or through analysing the variable importance of the variables). In this model you could include prior land use as well. Furthermore, some parts of the methods need clarification (specified below) and considering the scope of the manuscript section 3.3 should include more hands-on restoration management recommendations.
Minor comments
L16: What does ‘adjusted’mean in this context? Do you mean performed?
L16: It is unclear from this part in the abstract which independent variables were considered in the models.
L23: What do you mean with ‘recruitment attributes’? In case it is ‘recruitment density and diversity’ you could also consider to write it out, and not use attributes in the mauscript. In this way it is clearer what you refer to.
L71-73: This sentence seems to be composed of two sentences. Please rephrase or split the sentence up in two.
L89-93: I do not quite understand the purpose of this paragraph, as I suppose that the ypotheses are some sort of summary of the above mentioned text already.
L98: No need to mention the exact plot numbers here in the text or in the figure, as this is not so informative for the reader.
L100: The Kruskal-Wallis test is indeed just not significant, but with a p=0.06 it is close to significance. Therefore, I would like to suggest to take prior land use also into account in your models. You conclude that prior land use does not have an effect on recruitment (but see for example Jakovac et al., 2021 & Hordijk et al., 2024), but in my opinion it is not justified to conclude this when you do not test for land use in the linear models.
L167: Why is the 5km buffer a random factor in your models? Please expliani n your methods.
L251-257: You refer here to the intermediate disturbance hypothesis, but I do not see the connection with the presented results and the hypothesis. Could yo either elaborate on the connection or remove this paragraph?
L258: I would recommend you to describe the hands-on recommendations related to forest restoration and natural plant recruitment in this section. It would for example be interesting to understand based on your results where natural regeneration can be expected in planted forests and where additional management practices are needed.
L267: I would be careful with this statement if you didn’t include land use in your model.
L270: This is a reason to also look at the forest cover at 500 m radius surrounding the plots, as animal dispersal might occur at distances closer than 1 km radius (Schurr et al., 2009; Dent & Estrada-Villegas, 2021).
L304: Could you describe here the plot size?
L311: what is the reason that a height of 80 cm was chosen to measure the woody plants?
L341: What do you express with the first two axis of the PCA and why do you consider this information?
L352: You include several radii surrounding the plot in your study, but I would like to advise you to also take the 500 m radius into consideration. Especially animal dispersal can also occur at smaller radii (Schurr et al., 2009; Dent & Estrada-Villegas, 2021).
L364: What do you mean here with ‘adjusted’?
L365: What are the independent variables in the model?
L367: How can you perform a statistical analysis with a higher number of independent variable than observations? In that case you do not have any degrees of freedom left to trun the model.
L395: How does the final model looks like? Did you also perform a model with soil and landscape characteristics together? In that way you can analyse what the relative importance is of the different independent variables (either through comparing the standardized regression coefficients or through analysing the variable importance of the variables). In this model you could include prior land use as well.
Figure 1: A boxplot per land use type would probably be more informative.
Figure 2: Please give a more descriptive name to the dependent and independent variables (e.g. richness instead of D0).
References
Dent, D. H., & Estrada-Villegas, S. (2021). Uniting niche differentiation and dispersal limitation predicts tropical forest succession. In Trends in Ecology and Evolution (Vol. 36, Issue 8).
Hordijk, I., Poorter, L., Martínez‐Ramos, M., Bongers, F., Mendoza, R. D. L., Romero, P. J., ... & Meave, J. A. (2024). Land use legacies affect early tropical forest succession in Mexico. Applied Vegetation Science, 27(2), e12784.
Jakovac, C. C., Junqueira, A. B., Crouzeilles, R., Peña‐Claros, M., Mesquita, R. C., & Bongers, F. (2021). The role of land‐use history in driving successional pathways and its implications for the restoration of tropical forests. Biological Reviews, 96(4), 1114-1134.
Schurr, F. M., Spiegel, O., Steinitz, O., Trakhtenbrot, A., Tsoar, A., & Nathan, R. (2009). Long‐distance seed dispersal. Annual Plant Reviews Volume 38: Fruit Development and Seed Dispersal, 38, 204-237.
Author Response
Dear Reviewer,
Thank you for the careful and relevant revisions to our article “From stand to forest: woody plant recruitment in an Andean restoration project”. We made a careful evaluation for all of them. Please find our replies to the revisions below:

Reviewer 2 Report
Comments and Suggestions for Authors
This manuscript presents a comprehensive study of plant recruitment within a restoration project in Medellín, Colombia. The research effectively correlates various explanatory variables with plant diversity and density using appropriate statistical techniques. However, the manuscript requires significant improvements in its organization.
Many sections, particularly the introduction and discussion, are excessively lengthy and would benefit from streamlining. Additionally, there are minor errors resulting from literal translations from Spanish to English; while these do not detract from the overall quality of the information, I have marked them for correction.
Furthermore, the discussion section needs to be consolidated and currently lacks a “Conclusions” section, which is essential for summarizing the key findings of the study.

Please review the use of the comma. I explained the reasoning behind it in the manuscript. Also, consolidate the information, avoid the use of unnecessary expressions, use short sentences instead.
Author Response
Thank you for the careful and relevant revisions to our article “From stand to forest: woody plant recruitment in an Andean restoration project”. We made a careful evaluation for all of them. Please find our replies to the revisions below:
